# Radioactive Source Localisation via Projective Linear Reconstruction

**DOI:** 10.3390/s21030807

**Published:** 2021-01-26

**Authors:** Samuel R. White, Kieran T. Wood, Peter G. Martin, Dean T. Connor, Thomas B. Scott, David A. Megson-Smith

**Affiliations:** 1HH Wills Physics Laboratory, School of Physics, University of Bristol, Tyndall Avenue, Bristol BS8 1TL, UK; Peter.Martin@bristol.ac.uk (P.G.M.); Dean.Connor@bristol.ac.uk (D.T.C.); T.B.Scott@bristol.ac.uk (T.B.S.); David.Megson-Smith@bristol.ac.uk (D.A.M.-S.); 2Department of Aerospace Engineering, University of Bristol, Queens Building, University Walk, Bristol BS8 1TR, UK; Kieran.Wood@bristol.ac.uk

**Keywords:** radiation sensing, micro-gamma spectrometers, localisation, robotics sensing, radiation mapping, linear inversion, inverse problems

## Abstract

Radiation mapping, through the detection of ionising gamma-ray emissions, is an important technique used across the nuclear industry to characterise environments over a range of length scales. In complex scenarios, the precise localisation and activity of radiological sources becomes difficult to determine due to the inability to directly image gamma photon emissions. This is a result of the potentially unknown number of sources combined with uncertainties associated with the source-detector separation—causing an apparent ‘blurring’ of the as-detected radiation field relative to the true distribution. Accurate delimitation of distinct sources is important for decommissioning, waste processing, and homeland security. Therefore, methods for estimating the precise, ‘true’ solution from radiation mapping measurements are required. Herein is presented a computational method of enhanced radiological source localisation from scanning survey measurements conducted with a robotic arm. The procedure uses an experimentally derived Detector Response Function (DRF) to perform a randomised-Kaczmarz deconvolution from robotically acquired radiation field measurements. The performance of the process is assessed on radiation maps obtained from a series of emulated waste processing scenarios. The results demonstrate a Projective Linear Reconstruction (PLR) algorithm can successfully locate a series of point sources to within 2 cm of the true locations, corresponding to resolution enhancements of between 5× and 10×.

## 1. Introduction

### 1.1. The Nuclear Waste Problem

The volume of nuclear waste is ever-increasing and it is estimated that the UK will spend around £124 billion on nuclear decommissioning over the next 120 years. A significant portion of this will be attributed to the sorting of nuclear waste for volume reduction to ensure the most appropriate disposal route is adopted, based on an established waste hierarchy [1]. The UK’s Nuclear Decommissioning Authority has a radioactive waste strategy which requires that sort and segregation activities are carried out in order to “separate, on the basis of radiological, chemical and/or physical properties” [2]. The sorting process for mixed nuclear waste demands accurate radiation localisation and quantification. Currently in the UK, nuclear waste is being processed and stored in preparation for long-term storage in a deep Geological Disposal Facility. Therefore, undertaking such screening activities effectively and efficiently means that nuclear waste may be subsequently stored with confidence, underpinned by a detailed understanding of exactly what is contained within each waste package.

A defining feature of waste packages is their identified class category. In the UK, nuclear waste is distinguished into four distinct categories: Very Low Level Waste (VLLW), Low Level Waste (LLW), Intermediate Level Waste (ILW) and High Level Waste (HLW). Table 1 details the current, expected future arisings, and lifetime total radioactive waste volumes for each waste category [3].

The Box Encapsulation Plant at Sellafield Ltd. aims to accelerate this otherwise timely and inefficient process through the application of autonomous systems that employ the use of robotic manipulators [4]. This robotic system will act as a demonstrator for technology and processes capable of autonomously sorting and segregating assorted nuclear wastes. These operations are currently performed manually with humans operating the robots, but this leads to human error and excessive conservatism. The lack of precision means that personnel will often place lower levels of wastes into higher categories if uncertainty on the objects exists. For a robotic solution to autonomously carry out such a procedure, it would require detailed knowledge of the radiometric emitters, with a high spatial accuracy, enabling a robot arm to characterise, categorise and segregate objects correctly.

There are numerous gamma radiation detection techniques which can be used to both identify the presence of radiation and measure gamma energy (activity) through spectrometry. Scintillator crystals are commonly used for radiation detection and this spectrometry. The interaction of gamma radiation with these crystals results in a fluorescence which may be measured to determine the energy of the incident photon. This is a direct result of the band electronic structure. Upon impact, electrons from the valence band become excited, these excited electrons can migrate to either the conduction band, or the exciton band, located between the valence and conduction band. The result of this migration is the creation of a hole in the valence band. Fluorescence occurs when the electrons in the exciton band fall back to the valence band, emitting electromagnetic radiation in the visible spectrum in the process. It is this radiation which may be measured using optical techniques [5]. The development and commercialisation of solid-state micro-gamma-ray spectrometers, using semi-conductor and scintillating materials, has accelerated the current capability in radiological measurement on robotic systems.

### 1.2. Radiation Mapping

Systems able to connect positional information to radiometric data can locate radiation emitting materials, objects, or even areas of contaminated land. The outputs of these systems are commonly referred to as radiation maps. Radiation maps provide detailed information about the physical location of radioactive sources and are usually produced by interpolating between survey points to provide a continuous estimate (surface) of the radiation field.

Radiation mapping using robotic manipulator arms is reported by White et al. [6]. The team use a robotic manipulator with an attached radiation sensor package to accurately locate and 3D model groups of radioactive emitters on a flat surface. The radiation level of the emitters is reported in terms of intensity (activity) and surface dose rate. Selivanova et al. [7], also report the use of a robotic manipulator arm with an attached radiation sensor to make measurements of radioisotopes buried by concrete, steel, and water—making an assessment of the minimum detectable activity required to identify a given source. The isotropic nature of radiation detection means there is an inevitable blurring effect caused by the detection of radiation when the sensor is not directly facing the source and also due to the effect of other sources present. This makes it difficult to pinpoint where the sources are located. Precise localisation is an important factor to a sort and segregation system which uses robotic manipulators to grasp and move items. An example application is the sorting of Magnox swarf (the material produced after removing the cladding of the used fuel elements). The swarf contains cobalt-60 (Co-60), which comes from the Nimonic springs associated with the fuel cladding assemblies. While the proportion of Co-60 is comparatively small in relation to other elements, it is one of the most active constituents within the waste, hence, its precise identification and subsequent segregation is essential [8].

Methods for use within small scale waste sorting cells might be comparable to larger scale airborne radiometric surveys, which also seek to identify and categorise locations and types of radioactive material within a scene where the sensor is significantly separated from the surface. Aerial radiation mapping is now a common technique with both Boudergui et al. [9] and Ashley et al. [10] proposing it as a key tool for radiological emergency response. Connor et. al. [11,12] presents methods for surveying with Unmanned Aerial Vehicles (UAV) whereby position and height information is used to adjust the aerial measurements to improve the accuracy of the ground map. Gabrlik et al. [13] demonstrated a processing method for radiation mapping source localisation using UAVs, where the system used the inherent topology of the environment to vary the above ground flight altitudes to enhance the radiation map data. Ground vehicles or walking surveys ([14]) can overcome issues with source-sensor separation by significantly reducing the distances, however, this limits the area that can be surveyed in a timely manner.

### 1.3. Radioactive Source Localisation

There are several different devices available ‘off the shelf’ that demonstrate a higher precision radioactive source localisation. Lemaire et al. [15] demonstrated the radioactive localisation of sources using a gamma camera GAMPIX [16]. Higher spatial resolution was achieved using MATLAB post-processing with the GAMPIX running in Time over Threshold (ToT) mode. The ToT mode works by thresholding the pulses received at the charge sensitive pre-amplifiers on the Timepix chip. ToT mode can measure the time of the pulse in the threshold, which is directly proportional to the gamma-ray’s energy. Reference sizes of the pulses may be used to compare with received data to build a more accurate source localisation. Similarly, Paradiso et al. [17] implemented the use of a dual camera system—comprising of a gamma camera and a depth camera—to accurately localise radiation. This has now reached commercial availability as the iPIX camera system from Mirion technologies. However, it requires a ‘coded mask’ to be selected appropriately prior to operation according to the radiation intensity it will be used to locate. The coded mask works in a similar way to a pinhole camera, with an entrance plate in a shape created from an element opaque to radiation. This is placed over a photon detector which detects hits at different positions. Hence, it is able to identify hot-spots; however, the coded mask does not completely block gamma radiation in its covered sections, potentially skewing the radiation readings in the reconstruction algorithm [18].

The decommissioning sector in the UK, is beginning to use robotic arm systems for sort and segregation activities [4,6,7]. Hence, a thorough understanding of the characteristics of any gamma-ray detector used is of paramount importance. One key feature that must be qualified is the Detector Response Function (DRF), which describes how the detector records extraneous radiation measurements depending on its position relative to a radioactive emitter. Hence, characterising what the detector expects to see based on its location relative to a discrete gamma-ray emitter. Robotic manipulators are ideal platforms to characterise this 3D spatially varying collection efficiency of gamma-ray detectors, as they can position a detector around a radioactive source with a high spatial accuracy.

To this end, the use of robotic manipulators to determine the DRF for a micro-gamma spectrometer is demonstrated. The DRF is then used to apply a Projective Linear Reconstruction (PLR) algorithm to enhance the spatial resolution of the as-measured radiation maps. Such a technique has many potential radiation mapping applications on all systems from waste sorting cells to UAV-based surveys.

### 1.4. Micro-Gamma Spectrometry

Micro-gamma spectrometers are small devices which can be easily mounted on a range of robotic platforms, therefore enabling small-scale radiological recordings with geospatial information. This work uses a Kromek™ SIGMA-50 scintillator micro-gamma spectrometer [19]. This device (and other such devices) is routinely used for radiation mapping on robotic platforms as well as for handheld inspections. It is a Thallium-doped Caesium Iodide (CsI(Tl)) scintillator detector with the scintillating crystal sensitive to gamma-ray photons via the generation of an optical photon. The optical photons are converted to electrical pulses that are proportional to the energy and intensity of the incident gamma-ray photons. The crystal in the SIGMA-50 gamma spectrometer has dimensions of 25 mm × 25 mm × 50 mm and is located at one end of the detector behind a 1 mm thick aluminium casing.

### 1.5. Collimation

A common problem with current radiation mapping and detection systems is the low spatial resolution attained by such platforms. This can make it difficult to accurately locate radioactive sources, in particular when there are high levels of background radiation. This is problematic in decommissioning scenarios where it is important to identify and isolate the strongest radioactive emitters to reduce the final volume of consigned waste, and hence the overall cost. One method often used to increase spatial resolution is the addition of collimators. Collimators use dense materials (such as lead or tungsten) to artificially reduce the solid angle (field of view) of a detector by attenuating the incident photon flux to the sides of the device, leaving an aperture at the front to accept photons unimpeded. Adding shielding to the detector prevents extraneous gamma rays striking the detector thereby creating a preferential detection field of view. One key challenge, however, is that collimator materials do not block all gamma photons, but rather reduce the amount of gamma radiation reaching the detector by means of attenuation. The attenuation coefficient is the exponent which corresponds to the efficacy of a material (based on its atomic number and density) to stop incident gamma-photons by a combination of photoelectric absorption, pair production and internal scattering (both coherent and incoherent). Typically, with increasing photon energy the attenuation coefficient of a material decreases and so some gamma photons might reach the detector. Further to this, when such high-energy photons are attenuated, the associated scattering often produces many lower-energy photons which are also detected. This leads to numerous erroneous measurements within collimated detectors.

### 1.6. Resolving the Issue of Radioactive Source Localisation

Ionising radiation emitted from a point source is isotropic and follows an inverse square law. Some frequencies of electromagnetic radiation, for example the visible spectrum, can be focused using refractive and reflective optics. However, the refractive index decreases with increasing photon energy, meaning these traditional focusing optics are not applicable to gamma energies [20]. Therefore, an algorithmic method is presented instead.

The radiation detector may be thought of as a non-directional sum accumulator of gamma-rays and therefore a ‘counts’ reading may be represented as the summation of all the gamma-ray interactions within a time-period. If the emitting surface is quantised into a set of points, then any single measurement made by a detector placed above the surface will have contributions from all radioactive sources within sight. The relative proportion of the contributions will depend on the source strength, the distance, and the direction of intersection with the detector assembly. This representation can be formulated as a linear system Ax=b, where x is the unknown solution of the surface radiation emissions, A is a matrix which is represented by the DRF, and b is the measured response. Using this system, the ‘best solution’ of a scene containing one or more radioactive emitters in a radiation map, can be modelled. The problem then becomes one of mathematically obtaining an estimated solution to the linear system. The DRF approximation, experimental noise, and the over- or under-determined nature of the problem means it is not possible to solve the system analytically, therefore, an iterative approach based on the Kaczmarz method is used to estimate a solution. The Kaczmarz method is commonly used to solve systems of linear equations with uses in image reconstruction and tomography [21]. The general Kaczmarz equation, given in Equation (Equation 1), describes the process,
(1)$$x^{k+1}=x^{k}+\lambda \left(k\right)\frac{b_{i}-\langle a_{i},x^{k}\rangle }{∥a_{i}∥}a_{i}$$
where $x^{k}$ is the calculated solution for the *k*th iteration, $b_{i}$ is the measured radiation intensity at the *i*th position, $a_{i}$ is the DRF with the detector. λ is a relaxation parameter affecting the rate of convergence. It is hence possible to estimate the best solution, using a Kaczmarz method. Specifically, a randomised implementation of Kaczmarz deconvolution was applied for this study. This was selected over a conventional implementation due to its greater efficiency for this type of problem, reaching convergence more than 3 times faster [22].

## 2. Method

Initially, the DRF of the detector assembly was determined experimentally by measuring the detector efficiency as a function of 3D position relative to a small radioactive source. The experiment mapped the total intensity in multiple planes above a radioactive source using a KUKA KR 150 robotic manipulator. A photograph of the system is shown in Figure 1.

The active radioisotope scanned in this experiment was caesium-137 (Cs-137) with 36 kBq of activity, which was located within a sealed perspex enclosure. The perspex had a cylindrical shape, 50 mm in diameter and 50 mm in height. It was placed standing upright, in the centre of a 0.6 m × 0.6 m horizontal, flat area. The system was setup to scan across the surface in a raster pattern with a swath width of 1 cm. After each horizontal plane was completed by the scanning head, the robotic arm was raised vertically by 1 cm and the scan was exactly repeated at 10 different heights.

The detector was mounted within a lead (Pb) collimator with 6 mm wall thickness. The collimator did not protrude past the detector head plate. A diagrammatic view of the detection system is shown in Figure 1. This system was mounted on the end flange of the robotic manipulator. The gamma-ray counts and spectral data from the detector were recorded at a rate of 10 Hz and the robot arm was set to move at a predefined speed. The radiation measurements were synchronised with the robot arm position (accurate to the nearest 0.1 mm). Given the raster step size was set to 1 cm, it was logical to average the detector response data into 1 cm^3^ voxels.

### 2.1. Fitting the Detector Response Function

After collection of the experimental characterisation data, the random nature of radioactivity meant that the raw DRF included a degree of stochastic variation, as shown in Figure 2 (left). The raw measurements were used to fit a mathematical model of the detector. This DRF model was developed empirically to account for the two principal physical phenomena; the inverse square law and detector collimation. The inverse square law describes how the intensity of radiation emitted from a point source is proportional to $\frac{1}{R^{2}}$. The detector collimation is modelled using a complementary error response function (erfc) [23] shifted to transition from 1.0 (full detection) to 0.0 (no detection) at the collimation angle.

The final structure of the model was based on a product of erfc() and the inverse square law, combining these two phenomenon resulted in a model with two tuneable parameters: collimation angle and collimation gradient. The collimation gradient parameter allows for scattering and varying efficiencies at different photon energies around the collimator aperture. This model calculates the expected detector sensitivity for a source located at a position relative to the detector centre, as shown in Equation (Equation 2),
(2)$$\mathrm{DRF}\left(x,y,z\right)=\frac{\mathrm{erfc}(\epsilon \cdot [\theta (x,y,z)-\eta ])}{R{\left(x,y,z\right)}^{2}}$$
where DRF() is the Detector Response Function, *x*, *y*, *z*, are the spatial coordinates relative to the detector, erfc() is the complementary error function, θ is the angle between each point in space and the principle detection axis (the *z* axis in this instance), ϵ is the collimation gradient, η is the effective detector collimation angle, and *R* is the distance from any point to the detector.

To fit the detector parameters, the model was convolved with a 3D model of the source on the table and the result was compared with the raw DRF data shown. A least-squares optimisation procedure then adapted the ϵ and η values such that the modelled DRF accurately represented the raw DRF. The final result was an idealised DRF (Figure 2 (right)) free from measurement noise which was important for the stability of the iterative Kaczmarz deconvolution.

### 2.2. Kaczmarz Deconvolution Optimisation

The mathematical basis for Kaczmarz deconvolution is discussed in the theory section. However, Kaczmarz deconvolution assumes equal scan resolutions of the DRF and the input measurements. Consequently, to make the system as generalised as possible, a procedure was developed to generate a suitable Detector Response Function according to the scan resolution of the input dataset.

In the algorithm itself, first a measurement point is randomly selected, the region below it is then updated using a weighted projection of the DRF. The weighting co-efficient is the product of the relative error between the current solution and the measured value at the selected measurement point, and a dynamic relaxation parameter. The relaxation parameter adapted such that $\lambda =\frac{1}{10^{k/300000}}$, where *k* is the iteration number. In this way the algorithm initially prioritised convergence rate over stability, before gradually transitioning to prioritising convergence stability. The solution is updated in accordance with Equation (Equation 1) and this process is repeated until a manually predetermined number of iterations has passed.

### 2.3. Experimental Scenarios

In order to evaluate the algorithm four scenarios were set up to assess its capability. The radioactive materials used for testing were sealed within a cavity inside a short perspex cylinder (referred to herein as a ‘puck’, visible in Figure 1) and arranged on a flat surface. There are two types of of puck including Cs-137 and and Naturally Occurring Radioactive Material (NORM). The NORM pucks were comprised of crushed natural uranium ore (pitchblende). The source activity and location of the pucks is given in Table 2 and schematically plotted in the Appendix A for the four scenarios.

Scenario 1 was designed as simple test of the algorithm to resolve the location of two well spaced Cs-137 sources placed 30 cm apart. The second scenario used four pucks including a different and much weaker source material (NORM). The third scenario attempted to determine the spatial limitations of the PLR algorithm. This was achieved by determining the closest source placement the PLR algorithm could resolve. The two Cs-137 pucks used for scenario 1 were initially placed 15 cm apart and then moved progressively closer, in 1 cm incremental steps, until the limit was identified. The PLR algorithm did not resolve two distinct point sources at separations of less than 12 cm. The fourth scenario tested the ability of the PLR algorithm at resolving sources of different strengths. For this, seven pucks were placed randomly across the table. For each scenario the raster was either 1 or 2 cm. The *z*-offset is defined as distance from the top surface of the puck to the bottom face of the detector, this distance remained constant.

## 3. Results and Discussion

### 3.1. Scenario 1—Two Sources

The results of the scan and subsequent processing are presented in Figure 3. To visually present the effectiveness of the PLR algorithm, the data has also been initially processed using a simple and naive 2D linear interpolation. Figure 3a shows the approximate location of the radioactive sources used in the scenario, but their precise locations cannot be pinpointed. The data was subsequently input to the PLR algorithm, yielding the high spatial resolution result shown in Figure 3b.

This simple test indicates the PLR method can successfully locate the sources to within 1 cm, reducing the blurring effect measured by the system in the initial map. The PLR algorithm reports the sources positions at 15 ± 1 cm and 45 ± 1 cm on the *X* axis respectively and both 14 ± 1 cm on the *Y* axis, which is in agreement with their actual placement. It is possible that the accuracy is higher than 1 cm as when we consider the physical construction of the radioactive source pucks in use, they are of 5 cm diameter with an active internal cavity of 3 cm. However, the distribution of the active material within the cavity may not be uniform. If the active material were set to one side of the cavity, the PLR algorithm would only identify that particular side. This might explain the misalignment seen Figure 3b (right-hand puck), but there was no experimental ability to verify the internal distribution of material within the pucks.

### 3.2. Scenario 2—Four Lower Activity Sources

The results for scenario 2 are shown in Figure 4a,b for the simple interpolation and PLR processing respectively. The results demonstrate the algorithm accurately locates four similarly active, but relatively weak radiation sources to within 1 cm. Here the *X* positioning of the sources, left to right respectively is: 13 ± 1 cm, 42 ± 1 cm, 72 ± 1 cm and 98 ± 1 cm. On the *Y* axis, the positioning of the sources left to right is: 13 ± 1 cm, 14 ± 1 cm, 14 ± 1 cm, 15 ± 1 cm. It accomplishes this while preserving the relative source strength data in the process; however, the PLR algorithm does not recover the absolute source strength. It should be noted that the raw data collected for this map was of lower scan resolution (2 cm) than the first scenario.

### 3.3. Scenario 3—Proximity Limit

The third scenario attempted to determine the spatial limitations of the PLR algorithm. This was achieved by determining the closest source placement the PLR algorithm could resolve. After progressively reducing the separation of the pucks, a limit was identified where the PLR algorithm did not resolve two distinct point sources at separations of less than 12 cm. The interpolated raw data is shown in Figure 5a where it is not possible to identify that the radiation pattern is the result of two distinct separated sources. Interpretation of the data in this form could be ambiguous, in that this may be interpreted as a single elongated source. The PLR algorithm resolves the data into two distinct sources as shown in Figure 5b.

The white dashed circles indicate the independently measured positions of the radioactive sources. The identified positions of the sources on the *X*-axis, left to right respectively are 24.5 ± 1 cm and 36.5 ± 1 cm. The sources are less localised than scenarios 1 and 2, as the peak values are not included entirely within the white rings. This can be explained by weak radiological contributions from relatively distant sources skewing the local position of closer sources. This is ultimately a limitation of the method; however, it could potentially be improved with some prior knowledge of the source setup, or a higher resolution scan.

### 3.4. Scenario 4—Mixed Source Strengths

The fourth scenario tested the ability of the PLR algorithm at resolving sources of different strengths. For this, seven pucks were placed randomly across the table at the locations described in Table 2. In the interpolated raw data (Figure 6a), the sources are significantly blurred together.

The identified source positions are at co-ordinates (X,Y) cm respectively, as ordered in Table 2, Scenario 4: (73,44) ± 2 cm, (60,19) ± 2 cm, (40,19) ± 2 cm, (50,34) ± 2 cm, (44,46) ± 2 cm, (25,34) ± 2 cm and (17,19) ± 2 cm. The sources are only identifiable to within 2 cm in this scenario. This resolution is ascertained by measuring the distance from the centre of the expected source to the reconstructed centre point, for the least accurate source identified in Figure 6b. This reduction in accuracy is likely a result of the disparity in the source activities, ranging from 970 Bq to 36 kBq. It should be noted that there are a few artefacts present in the PLR algorithm result for both scenarios 3 and 4. These take the form of additional weak hot-spots which might be interpreted erroneously. These weak hot-spots can possibly be attributed noise in the measured raw data leading to the algorithm converging on a partially incorrect solution. To resolve this issue, the raw data set might be collected with greater detail (denser raster widths, or slower movement), therefore reducing the effect of random errors, or, with additional algorithmic checks to identify and remove erroneous results. The capability of the algorithm to recover the source locations in this scenario demonstrates the power of the PLR algorithm to analyse scenes with significant source activity differences.

### 3.5. Quantitative Improvement Measure

To quantitatively assess the effectiveness of the PLR algorithm a performance metric was developed by assessing a single line transect of the 2D data through the centre of the pucks for scenarios 1 and 2. The values for both the raw data and the best solution reconstruction are shown in Figure 7. A Gaussian distribution was fitted to each peak of both the interpolated and PLR processed data and the standard distribution is used to act as a measure of the quality of the localisation. A smaller value indicates a sharper fit, therefore indicating the effectiveness of the PLR method of reducing the blur inherent to the interpolated results. It is recognised that a Gaussian fit is a suitable approximation for our small round pucks, but less applicable for unusually shaped sources.

The standard distributions for the fitted Gaussian’s are presented in Table 3. An enhancement factor has been calculated for each puck as the ratio of values. We conclude that the enhancement ranges from approximately 5–10× with an average of 7.8× improvement. This quantitative analysis demonstrates that the PLR process improved the localisation and the overall positioning accuracy of radioactive sources scanned by a gamma-ray detector on a robotic arm.

## 4. Conclusions

The results of this paper demonstrate a post-processing technique, to aid the localisation and visualisation of radioactive sources measured by radiation mapping. Experimental testing using a robotic arm has validated the benefits of the PLR method for several multi-source scenarios. The algorithm demonstrates a capability of pinpointing radioactive sources to within 2 cm. Improvements between 5× and 10× were observed with a mean enhancement of 7.8×. In addition, a mathematical model is presented that approximates the Detector Response Function of a Kromek™ SIGMA-50, within a collimation assembly. This was fitted to experimentally measure data. The PLR algorithm then used an iterative randomised Kaczmarz method to compute best solution estimates.

The capability to accurately locate radiation on objects could be a powerful tool for a wide variety of nuclear decommissioning challenges. Although the focus of this research is on micro-gamma spectrometers on robotic manipulators at the sub-metre scale length, a PLR algorithm like this could be applied on many different radiation mapping platforms over different scale lengths (such as UAVs or ground vehicles). It is conceivable that the method could be applied in real time as part of an automated mapping platform. This would drastically reduce data collection and processing times whilst delivering fast and accurate radiation maps.

## Figures and Tables

**Figure 1 sensors-21-00807-f001:**
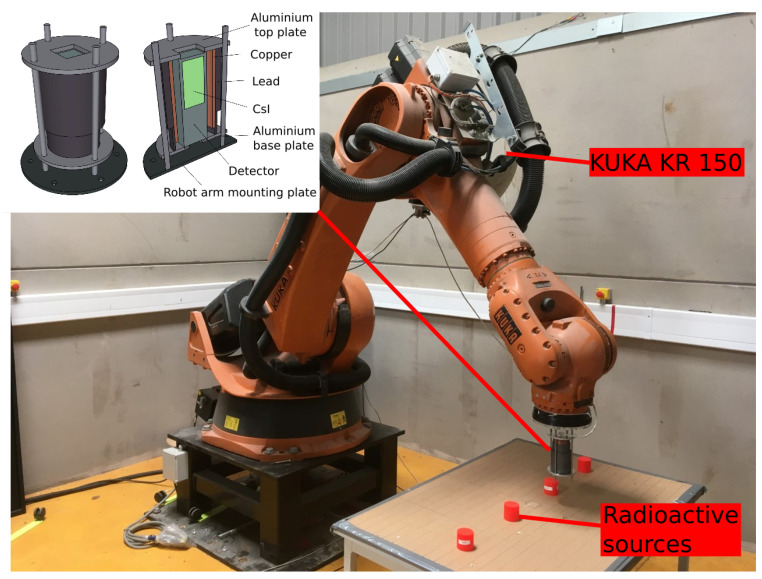
Photograph of the KUKA KR 150 system used, with a set of four sealed sources arranged on a table top. Inset—a detailed schematic of the detector and collimator setup.

**Figure 2 sensors-21-00807-f002:**
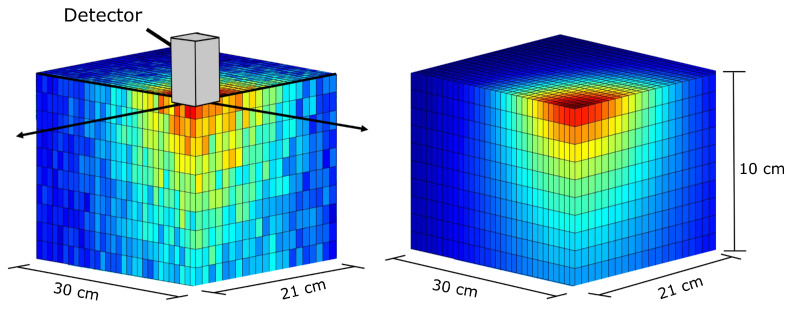
Quadrant cross-section of the as measured Detector Response Function (**left**) and the fitted model Detector Response Function (**right**). On the left-hand image, a model detector crystal (collimator and aluminium case not shown) is superimposed to highlight the experimental method. The detector crystal is outlined as an oblong shape with finite volume, although the algorithm assumed a perfect point, and all distances and angles are measured relative to the centre of the crystal.

**Figure 3 sensors-21-00807-f003:**
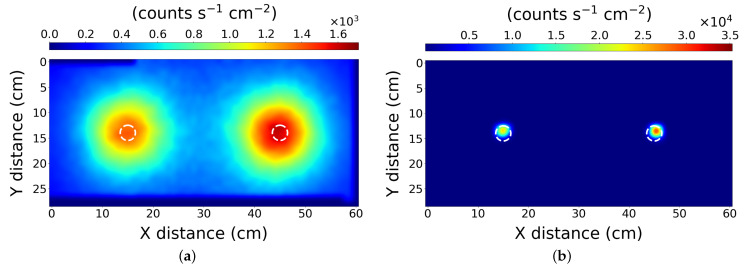
The results from scenario 1—Two similar sources. After processing with (**a**) simple interpolation, and (**b**) the PLR algorithm. The scan used a 1 cm resolution. The dashed white circles represent the true location of the source pucks.

**Figure 4 sensors-21-00807-f004:**
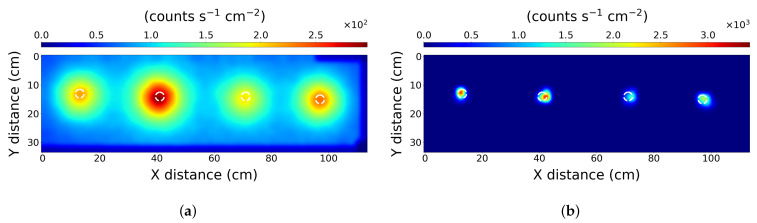
The results from scenario 2—Four lower activity sources. After processing with (**a**) simple interpolation, and (**b**) the PLR algorithm. The scan used a 2 cm resolution. The dashed white circles represent the true location of the source pucks.

**Figure 5 sensors-21-00807-f005:**
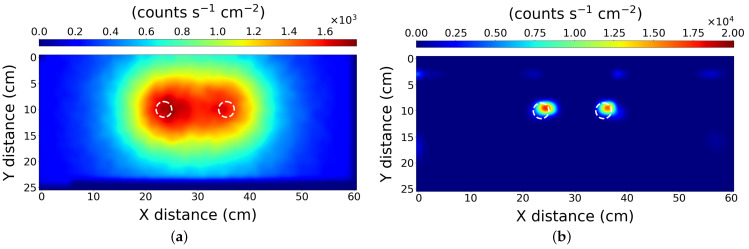
The results from scenario 3—Proximity limit. After processing with (**a**) simple interpolation, and (**b**) the PLR algorithm. The scan used a 1 cm resolution. The dashed white circles represent the true location of the source pucks.

**Figure 6 sensors-21-00807-f006:**
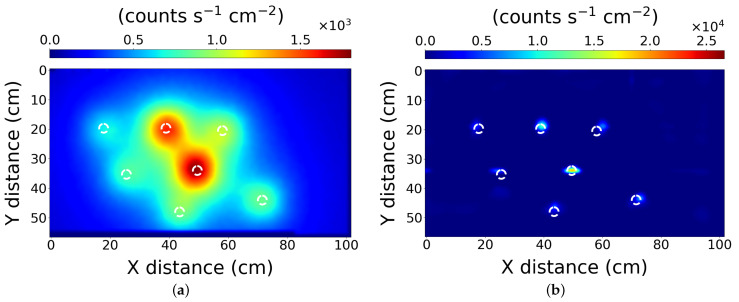
The results from scenario 4—Mixed source strengths. After processing with (**a**) simple interpolation, and (**b**) the PLR algorithm. The scan used a 1 cm resolution. The dashed white circles represent the true location of the source pucks.

**Figure 7 sensors-21-00807-f007:**
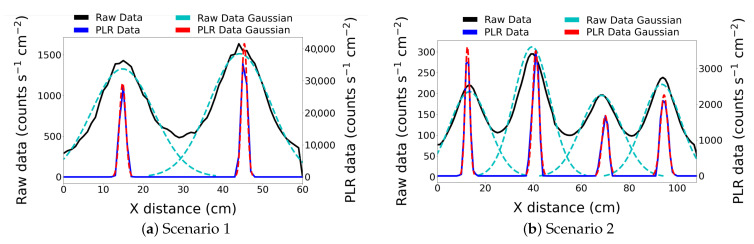
(**a**) Scenario 1 and (**b**) Scenario 2. Comparison of 1D transects through the interpolated raw measurements (Raw Data) against the best solution estimated by the linear reconstruction technique (PLR data). Best fit Gaussian distributions have been fitted to both to enable a quantitative resolution assessment to be made.

**Table 1 sensors-21-00807-t001:** Current, future arisings and lifetime total expected volumes of different nuclear waste categories. All values are a direct reproduction from the Nuclear Decommissioning Authority report [3]. * This negative value reflects the future conditioning of waste volumes.

	Volume (m${}^{3}$)		
Waste Category	Reported (as of 1 April 2019)	Estimated Future Arisings	Lifetime Total
HLW (>10 GBq/kg)	2150	−760 *	1390
ILW(<10 GBq/kg)	102,000	145,000	247,000
LLW (<12 MBq/kg)	27,400	1,450,000	1,480,000
VLLW (<100 kBq/kg)	1040	2,830,000	2,830,000
Total	133,000	4,420,000	4,560,000

**Table 2 sensors-21-00807-t002:** Puck types and placements relative to the robot base coordinates.

	Type	Activity (kBq)	*X* Pos. (cm)	*Y* Pos. (cm)
Scenario 1	Cs-137	31.0	15.0	14.0
	Cs-137	36.0	45.0	14.0
Scenario 2	NORM	2.9	13.0	12.5
	NORM	4.7	41.0	14.0
	NORM	2.7	71.0	14.0
	NORM	3.0	97.0	14.5
Scenario 3	Cs-137	36.0	23.5	10.0
	Cs-137	31.0	35.5	10.0
Scenario 4	NORM	6.11	71.5	44.0
	NORM	1.38	58.0	20.5
	Cs-137	31.0	38.9	19.7
	Cs-137	36.0	49.5	34.0
	NORM	1.21	43.5	48.0
	NORM	3.49	25.5	35.3
	NORM	0.97	17.8	19.7

**Table 3 sensors-21-00807-t003:** The standard deviations of the fitted Gaussians as in Figure 7.

	Source Number (Left to Right)	Standard Deviation Raw Data (cm)	Standard Deviation PLR Data (cm)	Enhancement
Scenario 1	1	7.8 ± 0.2	0.78 ± 0.01	10
	2	7.6 ± 0.2	0.70 ± 0.01	10.9
Scenario 2	1	8.3 ± 0.3	0.98 ± 0.01	8.5
	2	7.5 ± 0.2	1.35 ± 0.01	5.6
	3	8.6 ± 0.3	1.37 ± 0.05	6.3
	4	8.0 ± 0.3	1.55 ± 0.04	5.2

## Data Availability

The data presented in this study are openly available at https://data.mendeley.com/datasets/5w5tr7xdrr/1.

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
