# Peer review of "Radioactive Source Localisation via Projective Linear Reconstruction"

_sensors, 2021, doi:10.3390/s21030807_

Round 1

Reviewer 1 Report

Good piece of work - recommend it be accepted forthwith.

Author Response

Thank you very much for your recommendation for publishing. In response to comments and suggestions from our other reviewer, changes have been made to improve our manuscript, which we would like to make you aware of:

A paragraph on the mechanisms behind scintillator detectors has been added to aid the general readership, as you recommended. Please find this on lines 46-58.

A greater emphasis has been placed on the application of radiation mapping using robot arms and the limitations thereof. Please find this addition on lines 56-80. The airbourne survey section has been reduced in length, please see lines 81-92 for update.

In section 2.1, additional explanation has been given to explain the modelling process behind the DRF.

The description of the scenario setup (which was overly verbose) has been substantially reduced and the figures of data before and after processing have been placed side by side for visual impact – it makes the data much easier to interpret. A new section “3.5. Quantitative improvement measure”, was introduced to concisely discuss the Gaussian fitting of the line plots.

Reviewer 2 Report

I find the technical content of this article to be good and I'm not fully opposed to accepting it as is, but I think a few minor changes would greatly improve the impact.

First, it could generally use a bit more editing. Overall it's fine, but there are some typos and awkward phrasings present, especially in the second half. I recommend the authors do another round or two of proofreading. Some of the text is a bit verbose, which makes portions tedious to read and it could use a little trimming to cut down on the overall length slightly.

My background is in radiation detection so this article is fine for me, but I think the addition of some brief overview (no more than 2 or 3 paragraphs) or at least references on the basics of radiation detection, especially scintillators, and maybe also radiation imaging might help make this more accessible to the general audience of this journal. Preferably this would appear early on in section 1 before getting into some of the details of the detector system and so forth.

More emphasis should also be placed in the introduction on the specific application the authors are interested in, i.e. imaging/mapping with a robotic arm, and the problems/limitations with the existing or naive approaches that motivated the research. This is addressed more in the presentation of the results, but I feel it could be better articulated up front as well to provide readers with more context before diving into the details. Perhaps this could replace the portion of the literature survey dealing with airborne radiometric surveys? I feel like that topic is not really that relevant to small scale raster radiation mapping with a robotic arm, which probably has better parallels with radiation imaging techniques like say scatter cameras.

Section 2.1 - please add a reference or explanation for the choice of model for the DRF. I know this is basically a standard choice, but a broader audience will likely be confused.

I would prefer to see the results from each of the test scenarios paired with their descriptions rather than in two separate sections. It is a bit difficult to keep the relevant details in mind with them separated, making it harder to read.

Page 10 - please don't refer to the standard deviation of the Gaussian fits as "sigmas", call them "standard deviations." Same anywhere else.

A minor nitpick, but I would prefer the authors avoid the term "ground truth" in the way that they use it. Ground truth is a reference value that is provided independently or at least quasi-independently to benchmark against. I would prefer something simple like "solution" or "best prediction" but I'll leave it to the authors' discretion.

Overall a nice article that is mostly well presented as is, and with just a bit more polish I think it will make for a nice contribution.

Author Response

Thank you very much for your review. Your feedback has been instrumental in helping us to improve this paper.

The paper has since undergone a few iterations of proof reading by all authors, so we hope that the new version of the manuscript is of higher quality, greater clarity and containing fewer errors. A few steps have been taken to make the language more concise, as detailed.

A paragraph on the mechanisms behind scintillator detectors has been added to aid the general readership, as you recommended. Please find this on lines 46-58.

A greater emphasis has been placed on the application of radiation mapping using robot arms and the limitations thereof. Please find this addition on lines 56-80. The airbourne survey section has been reduced, however it has been kept in, as we believe that this algorithm has a broader application which could be useful to the airbourne mapping community (we have a paper coming out for this shortly). The revised paragraph can be found on lines 81-92.

In section 2.1, additional explanation has been given to explain the modelling process behind the DRF. I hope that this reads with improved clarity now.

In light of the scenario recommendation, the results section has been modified to increase understanding and clarity. The verbosity of our description of the scenario setup has been substantially reduced and the figures of data before and after processing have been placed side by side for visual impact and enhanced ease of interpretation. A new section “3.5. Quantitative improvement measure”, was introduced to concisely discuss the Gaussian fitting of the line plots.

All references to “sigmas” have been removed and replaced with “standard deviation”

The word “Ground truth” has been replaced with “best solution”

I hope that you find the revised paper to be more polished. Thank you for your valuable input.

This manuscript is a resubmission of an earlier submission. The following is a list of the peer review reports and author responses from that submission.

Round 1

Reviewer 1 Report

Dear authors,

it has been interesting reading your paper and the main goal of the article has been reached.

I have some comments (notes in the attached file) mostly regarding the results:

  • A schematic representation of the two scenarios could help the experiments.
  • in the picture n2, I personally do not draw the detector like that, but only its center.
  • it is not clear the de-convolution method applied to find the source position. Is it the centroid of the distribution? do you take into account the sigma of the distribution? Is it reported?
  • it could be interesting to report the results of the two scenarios interm of height and scan speed. Or in other words, are the results dependent on the scan velocity and heights?
  • The results should be presented between the measurments, the real position of the sources and the PLR algorithm, with all the uncertanties.
  • The figure n5  gives only a rough knowledge of the comparison. Why not indicating the difference (weighted with the sigma) of the two source localization?
  • Fig.7: The white dashed circles are not clearly visible.
    It is not clear how the SRF takes into account the sigma in the distribution.
  • Fig.8: it seems to me a disalignment on the left, between the PLR and the pucks. Only a visualization problem?
  • Conclusion: could you please discuss the resolution 0.5 cm better?
    0.5 cm?
  • Line 293: In my experience, the UAV community do not have the 0.5 cm knowledge typically as a key constraint.

Reviewer 2 Report

Authors describe method for pinpointing radiation sources using projective linear reconstruction based on Kaczmarz deconvolution. The topic is important and interesting for the readers. Paper is well written and scientifically sound, but some editorial work would be necessary before publication.

Authors should correct:

  • There are lowercase "x" instead of proper multiplication mark "×".
  • Physical units should not be in italics.
  • There are dots "." instead of proper multiplication sign "·". On lines 177 and 231, there is no multiplication sign whatsoever.
  • There should be blank space between number and unit (line 183, 190,...).
  • erfc in eq. (2) should not be in italics.

Reviewer 3 Report

This is obviously a well written paper which presents the results achieved clearly and gives a general overview of the research landscape in this area.

However my big complaint is that the two scenarios presented are too simple and the challenge presented to the developed algorithm / system is not great enough. Two Cs-137 sources a clear distance apart, and then 4 sources a clear distance apart doesn't feel like a significant challenge to me. I'd like to see a greater challenge with perhaps sources very close together, sources several orders of magnitude greater than the other etc...

Specific comments:

Line 31: You say Five categories of waste and then list only three. I think an error is that you have listed Very Low Level Waste as LLW (should be VLLW), but even if this is sorted then that is four - not sure what number five is.

Line 128: I would appreciate a good diagram of the detector so as to ascertain whereabouts within the enclosure the crystal actually is. We used the RadAngel detector for example and the exact position of the CZT crystal within the enclosure dramatically changed our results.

Line 161: You mentioned the "Under/Over determined nature of measurements", does this refer to the uncertainty in measurements generally? I was confused as to what this meant - could this be explained more?

Line 171: "Reaching convergence more than 3 times faster", three times faster than what? I was a bit unsure.

Line 231: The Cs-137 sources are mentioned in terms of Bequerels which makes sense. However, the NORM sources are in terms of Bq.g^-1. This seems odd to me as I am not sure why these are not in Bq too. 5g of a 20 bq.g^1 will obviously give more signal than 2g of a 30 bq.g^1 source - so why not give the overall activity of each puck rather than activity per gram?